# Proof of Concept (PoC) 1.0—Implementing a Bioshading System Design Method

**DOI:** 10.3390/biomimetics6010008

**Published:** 2021-01-19

**Authors:** Maria João de Oliveira, Vasco Moreira Rato, Carla Leitão

**Affiliations:** 1DINAMIA’CET_Iscte, ISCTE-Instituto Universitário de Lisboa, 1649-026 Lisboa, Portugal; 2ISTAR_Iscte, ISCTE-Instituto Universitário de Lisboa, 1649-026 Lisboa, Portugal; vasco.rato@iscte-iul.pt; 3Rensselaer Polytechnic Institute, Troy, NY 12180, USA; leitac@rpi.edu

**Keywords:** biomimetics, design methodology, plants adaptations, shading systems

## Abstract

Nature provides a remarkable database of possible adaptation strategies that can be implemented in biomimetic design of shading systems. However, at this moment, successful design methods are conditioned to a limited knowledge and ability to emulate nature’s strategies to meet corresponding functional needs. The implementation of biomimetic processes has some major challenges: (1) the search and selection among several databases of appropriate strategies adopted by nature; (2) difficulties in reading, interpreting and translating at different scales; (3) connection problems between concepts and material premises. The selection of nature models is a very common situation among architectural projects. Proof of Concept (PoC) 1.0 was the first experience of application of the Bioshading System Design Method (BSDM). BSDM is a problem-based method that guides its users since the initial architectural challenge definition, improving users’ capabilities to interpret and translate nature strategies into architecture design, until its final state of creation, it’s physical condition. This experience enabled us to validate and evolve initial decisions, based on users experience and evaluation. At the end, PoC 1.0 revealed to be a fundamental step into the final version of BSDM.

## 1. Introduction

In 1917, D’Arcy Thompson, in his On Growth and Form, established the theoretical problematic of design, conceptualizing that the evolution of form over time is based on an initial structural pattern [1,2]. The relationship between form and environment, the conception of evolutionary design as an evolutive pattern, and the limitations of technology as generative and evolutionary processes were some of the fundamental issues for the development of nature-based theories and practices into contemporary design. More than morphological studies of shape and structures, Thompson’s work launched the basis for a clear understanding of the growth and adaptation of form in specific site conditions in what is known as form-finding processes [3]. There are innumerous aesthetic ideas and formal references in art and architecture inspired by Nature. From Sullivan’s (1856–1924) ‘Golden Door’ organic ornamentation and Wright’s (1867–1959) organic architecture praxis, to Gaudí (1852–1926) catenary curve models made with weighted hanging chains, wire and rope, exploring and studying structural processes, and Otto (1925–2015) and the Institute of Light-weight Structures (ILS) experiments on structures and gravity using analogical form-finding models. The precedent mentioned works could be considered as the ‘classical’ basis for computational models of form-finding.

More than reproducing a natural form, a natural morphological pattern that creates organic architectural shapes, new emerging methods and generative design theories look into Nature as a set of processes, described through the employment of scripting and coding techniques, which in turn describe and reproduce relationships between a variety of systems, not only natural but also artificial ones. This link between mathematical models-based evolutionary adaptations, informed by the natural environment, is producing a naturalizing-architecture. First used by Frèdèric Migayrou and Marie-Ang Brayer in 2013, Naturalizing-Architecture is a term that derives from our ability to digitally model and fabricate based on similar phenomena of the natural environment. To Migayrou (2003), the creation of representative models of natural complexity of growth is among the main issues of natural digital modeling of form and structures [4]. In the same year of 2013, Oxman goes further, arguing that today’s architecture requires an informed process that encompasses a model that should link analysis and synthesis, performance and generation, tectonic integration of form, structure and material. Oxman’s informed processes integrate four phases: (1) the formation of the process (parametric design) is sustained by mathematics, geometry and topology; (2) the performative component (performance-based), supported by analysis and synthesis; (3) Generative techniques; and (4) Fabrication tools. Parametric design is a mathematical model of shape forming, which can be defined as a topological differentiation process based on computational models of associative geometry. The impact of a given environmental context on the sustainability and efficiency of a project should be considered even during its formulation phase. Performance-based design is achieved when computational analysis and simulation are integrated with the generative design process [5,6]. Generative evolutionary processes require the understanding of design as an algorithmic process. Fabrication requires material, assembly and construction strategies, as well as expertise in order to master this design field [3].

This paper is based on a research that proposed a Bioshading System Design Method (BSDM) construction process, developed on a problem-based approach using terrestrial plants as inspiration [7]. Starting with the architectural challenge of design, solutions will be sought in Nature to solve specific Bioshading systems performance requirements. The hypothesis that sustains the method development lies over an informed process that integrates and interrelates three domain areas: 1-Architecture; 2-Nature; and 3-Artifact. In this context, the Architecture domain roots its basis on the formation of the process, computational environmental analysis and diagnosis. This formation process is conducted through environmental analysis software, integrated through parametric design tools. The Nature domain is defined through an abstraction process. Sustained by a plants mapping process matrix, the creation of a meme’s semantics triggers a performance-based design process, which is achieved when computational analysis and digital simulation are integrated with the exploration of shape and structure through generative design processes. The Artifact domain is the physical materialization of the design concept that enables its evaluation and emulation. Performance-based design processes and digital fabrication tools are integrated components, supporting the creation of the artifact. BSDM is intended at both academics and professionals. In this perspective, the method is supported by a digital toolkit. The idea is that the toolkit allows greater proximity between the users and the process, working as a pedagogical vehicle of information, promoting debate between working groups, and facilitating the development and organization of the different tasks to be carried out during the process. This paper describes a real-time proof of concept (named PoC 1.0) that was ideated and implemented to validate the preliminary version of BSDM. Its results led to the elaboration of the final version of the BSDM and its toolkit components.

## 2. Method

PoC 1.0 was conducted through two separate four-hour sessions. Ten voluntary participants, students and professionals of the architectural field, organized in pairs, carried out this experience. None of the participants had knowledge or base experience in the field of biomimetics (Figure 1). For this purpose, a computer laboratory was used.

The main goal of this experience was for participants to be able to develop a façade shading system to a pre-determined building and defined context, using the Bioshading System Design Method, version 1.0. PoC 1.0′s sessions aimed at testing and evaluating the method considering three criteria: (1) Method Clarity (evaluated by the participants at the end of the experience); (2) Efficiency and effectiveness of the PoC 1.0 sessions themselves (participants were invited to evaluate (i) the clarity of the oral presentation on the method, and the supplied digital material regarding the method, (ii) duration of each session, and (iii) the available means, as computers and software); (3) Method Operability and its Outputs (evaluation performed by the team involved in the development of the method, about each project output from these sessions, their method’s clarity and applicability, goal definition, biomimetic meme generation matrix, design solutions and its technical implementation). 

Bioshading System Design Method (BSDM) relies on a circular process ordered in nine phases, equally distributed by three domains: Architecture, Nature and Artifact (Figure 2). Initiating its journey with the Architectural domain, this new method has a problem-based concept design approach. The first session of the PoC 1.0 experiment guided the participants through the Architecture and Nature domain phases, in order to, respectively, define the shading system goals and create its concept design Biomeme. The Goals definition consisted of determining the main functions (performance requirements) of the shading system to be designed, as well as the actions that would support them and the agents that would enable it (Table 1). As BSDM is based on terrestrial plants, the Biomeme is the result of the creation of a fictional meme as a product of the study of a given plant, combined with the aimed functions of the shading system (Table 2). The second session was essentially focused on the Generation and Simulation phases of the Artifact domain, and it was strongly devoted to the digital design project, considering types of structure, actuation, fabrication and materials.

The experience was conducted through a defined time script (Figure 3) having a digital kit as support. The digital kit was composed of several folders containing: (i) the digital 3D model of the case study building and its context surroundings, (ii) the Climate Consultant 6.0 graphic analysis [8] of Lisbon’s climate, (iii) Ladybug [9] graphical analysis of the case study building’s south façade, (iv) tables and diagrams containing shading façade essential functions, actions and agents, (v) a list containing several terrestrial plant types and adaptation strategies, (vi) a Biomimetic meme path matrix diagram in order to help the participants define its fictional Biomeme, and (vii) two tables listing the main types of structure and actuation of shading systems.

The first PoC 1.0 session opened with a 30 min introduction of biomimetics and architecture. It was a chronological presentation aimed at contextualizing the application of biomimetic values and principles not only in architecture but also in other relevant fields as mechanics, design and materials science. The relationship between architecture and terrestrial vascular plants was pointed out as a case study and as an inspirational motto, and its link was justified based on plants’ and buildings’ similar physical condition. Finally, a brief presentation of the most used design, simulation and analysis tools, as well as computer aided manufacturing (CAM) resources currently available to architects, were also presented and discussed, identifying its strengths and weaknesses, throughout the architectural design process.

Entering the Architectural domain, at the Identification phase, PoC participants were presented to the case study building and its context. The selected case study building integrates a proposal for a students’ residency program, which also houses coworking and services spaces. Located in Lisbon, inside the Cidade Universitária Campus, the analysis target was the south façade of the building. A three-dimensional model of the Cidade Universitária Campus was given to the participants. A complete climate analysis of the city of Lisbon—using Climate Consultant 6.0 (CC) software [8]—was explained to the participants, containing graphical representations of annual temperatures, irradiation, daylight and wind velocity, and solar shading and psychrometric charts. In the second phase, Analysis, participants were introduced to Ladybug analysis charts and diagrams. Based on parametric information, Ladybug can perform real-time analysis, providing the possibility to extract two or three-dimensional diagrams, schemes and charts into/over the three-dimensional model. Dry bulb temperature, irradiation, total direct and diffuse radiation, urban shade benefit, shading comfort façade design, wind speed and air temperature roses were the diagrams and charts that were provided to the participants. A process of interpretation and analysis was then conducted. After a context and climatic analysis, participants were invited to Diagnose, defining which should be, in their perspective, the shading system’s main functions for that case study façade. Three base tables (Table 2) were supplied, containing (i) the shading system’s main functions, (ii) pointing some of the most relevant shading systems actions, and (iii) enumerating some of the agents that could trigger these actions. During the Diagnose phase, participants started working in pairs, which triggered some effective discussions over their conceptual aspects regarding the shading system pairs would later propose. From this brainstorm, the five groups were able to define their shading system’s main goals, as well as their functions > actions > agents semantic relationship, achieving at the end of this phase the so-called Challenge definition.

The second part of the first PoC 1.0 session was all about the Nature domain. The facilitator presented and explained the terrestrial plants’ vascular system, its relevance and main functional organs and features. During the presentation, several analogies between plants events and features, and the man-built environment functions were a major contribution to initiate an individual and creative link between the natural and the humanmade systems. To engage the working groups at the Discover and Exploration phases, an introduction was made on plants’ adaptation strategies -morphological, physiological and behavioral-, in order to explain how to use the supplied plant adaptation data matrix in its digital format, as well as how and where to search for the presented data or search for other adaptation events (fundamental online resources such AskNature, Biomimicry 3.8, Basic Biology, among others). It was then necessary to clarify the creation process of the Biomeme. From the several available surveys, each group was invited to engage in the Exploration phase to elaborate a Meme event table (Table 3), where they selected the plant adaptation events that could resemble their shading system’s defined functions.

In order to dissect the selected meme events, participants stratified those events according to its type of adaptation, strategy, main principles and features. This stratification is essential for the user of the method, allowing not only to extract the several characteristics and properties of each of the selected memes, as well as transport them through interpretation to the architectural lexicon. Adaptation and strategy will enable the meme categorization in the fields of its actuation. Principles are the BSDM user’s first approach to an individual interpretation of the meme event, while features are the pattern, material and performative characteristic observed by the BSDM user in that specific meme. After completing this task, the groups were ready to Conceptualize their Biomeme. The Biomeme conceptualization was produced with the completion of the Biomimetic Meme path matrix (Table 3). In PoC 1.0, the Biomimetic Meme path matrix crossed the shading system’s main functions with the selected meme events. Extracted from the previous meme events table, and in addition to the functions selected for the shading system, the Biomimetic Meme path matrix also crossed other inputs such as adaptation and strategy types, pattern, material and performative features. Through this process, each group achieved its Biomeme that puts together all the events whose occurrence is in majority.

After a one-day reflection gap between sessions, PoC 1.0’s second session was entirely devoted to the Artifact domain. The session was initiated by an oral presentation about the types of structures, mechanisms and actuations of shading system. An oral debate was encouraged in order to promote brainstorming between the groups. Two digital documents were supplied to the participants, containing synthesized information about shading systems structural types and possible types of actuation. The shading system types of structure document contained a short description, pros and cons of the each type of structure and possible actuation clues for its implementation. The actuation types document also contained a brief description of the actuation, its pros and cons and some required resources and knowledge for its implementation. The following period was completely devoted to the groups’ shading systems design. As it is represented in the PoC 1.0 time script, one hour of the second session was programmed to be dedicated to the Simulation phase; however, participants required to use it for the Generate phase design process. The last 40 min of this session was intended at hearing the PoC 1.0 participants’ opinions about the experience, and for them to evaluate the Method Clarity. From the PoC 1.0, five different projects, with different levels of development, emerged. These will be hereafter designated by letters A, B, C, D and E. A description of the groups’ results is presented in next section.

## 3. Results

### 3.1. Group A (Luísa Almeida and Ana Castanho)

Group A’s shading system’s selected functions were related to the system’s ability to block/let pass the direct solar radiation, enabling a convenient and constant external view connection, ensuring the building’s natural ventilation. In order to achieve these performance functions, the selected actions were permeability, intersection, material and scale. Their idea was to design a system that could be either permeable or opaque to light in different moments of the day. Materials would perform a significant role during this action, while scale was the key action that would enable natural ventilation. Their system was rooted in four fundamental agents: density, scale, pigment and pattern. Density was based on the repetition of the same element at different scales; pigment was related to the visual permeability of elements but also with the chromatic composition of the façade; and pattern composition was linked to the form and motion of the elements of the shading system. During the Nature domain, the group’s selected memes were bioluminescence, epidermis, nyctinastic movements and vernations. Their fictional Biomeme (Table 4) was a system with permeable/opaque ability, scale variations, with dynamic strategies and behavioral adaptation abilities, that should be materialized through a multilayer and perforated system, using porous material properties and open/close mechanisms.

During the Generate phase, a triple-layered façade was designed, composed of bi-directional radial foldable elements, organized by three different scales (Figure 4). Different scale elements were arranged in the three layers façade, producing several overlapping areas in the final composition. The foldable elements were composed of triangular frames, coated by two different materials. When rotated clockwise, the elements exhibit a perforated textile; when rotated counter-clockwise, the elements exhibit an opaque textile. The system was conceived to respond to the sunlight position automatically.

### 3.2. Group B (Susana Neves and João Parcelas)

After a careful analysis of the CC and Ladybug climate analysis, Group B considered that the most relevant, case-study-related functions for the shading system should privilege the external views, natural ventilation and convenient architectural integration. It is important to refer that these functions are always rooted in the context of controlling solar radiation. Their selected actions were permeability and material. Permeability enables the connection between interior and exterior, natural ventilation, while material opens a wide range of possibilities for the shading system’s performance and to its proper architectural integration. In order to perform the selected actions, morphology and opacity were the elected agents. Morphology enables the creation of material/structural/motion integration, and adds a new layer to the material action. Having created a very flexible combination of elements during the Architecture domain, the group needed to be more accurate during the Nature domain phases. The selected memes were trichomes, nyctinastic movements and diaheliotropism. The Biomeme (Table 5) privileged external views, natural ventilation and architectural integration, using dynamic strategy and behavioral adaptation through an adaptive pattern composed of flexible material with tracking features.

The proposed solution was a stretched and bent vertical system composed of perforated and translucent flexible materials (Figure 5). The façade is cladded with vertical strips (all of which with the same width), and the system works in one stretch and bend consequent loop. A sun-tracking system controls this loop. When direct sunlight needs to be blocked, the translucent material is stretched and the perforated material is bent; when sunlight is an advantage, the perforated material is exposed and the translucent material is bent.

### 3.3. Group C (Raquel Martins and Carlos Sequeira)

Direct radiation, diffuse radiation, glare control and natural ventilation were the main functions considered by Group C. In order to perform these functions, the selected actions were permeability, material and scale. The selected agents were density, pattern, opacity and structure. Permeability is achieved by density and pattern, material by the opacity and scale through structure. This was a group that dedicated some quality time to the Discover and Exploration phases, during the period of the Nature domain. The selected memes were canopy plants, endothermic, xylem, superhydrophillic, phloem and bioluminescence. The created Biomeme (Table 6) aimed at controlling direct and diffuse radiation, glare control, and to ensure natural ventilation through the façade. The intended strategy was dynamic, with physiological adaptations. Features should resemble a multilayer bubbles pattern, using flexible and/or sponge materials, enabling unidirectional movements and storage behavior.

The fundamental idea was to create a living curtain façade that, using solar radiation, could heat collected rainwater for domestic use. Thus, the project was composed of double-layered vertical pipes, punctuated by double-skin rubber spheres equipped with individual heat sensors (Figure 6). Every time the sensor identifies direct solar radiation on a bubble, stored water enters the pipes and inflates the rubber bubble that, using solar radiation, heats the water. On the other hand, inflated bubbles create a physical and visual barrier between the internal and external environments (Figure 7). Furthermore, as the pattern is spherical, there is no total ‘blackout’, and ventilation is assured. During the night, with no sensing of solar radiation, the deflated bubbles glow, creating an iconic lighting effect (the glow process was intended to be produced by the material properties—but the theme was not developed).

### 3.4. Group D (Diana Gabão and João Sousa)

Based on comfort requirements, climate and urban surroundings, Group D defined that the façade has to block direct solar radiation, assure external views, promote natural ventilation and is appropriately integrated in the building’s architecture. In order to achieve these functions, selected actions were permeability, reflection, refraction and material. Since a very early stage, material was an important factor due to its potential to transform and achieve almost infinite viable combinations. To perform the aforementioned actions, the selected agents were translucency, morphology, pattern and density. At the Nature domain, the group focused on morphology, pattern and density actions, and selected diaheliotropism, endothermic, asymmetry, stoma and epidermis as their memes. Their Biomeme (Table 7) aimed at responding to direct radiation, maintaining a connection to external views, still considering the architectural integration. In order to design the shading system, a static strategy was conceived using morphological and physiological adaptations. The shading system operates through a multilayer pattern, composed of hard material, with concentric movements and/or cellular performance features.

The design of this group’s shading system was based on a main hexagonal grid, sub-divided in triangular parts (Figure 8). This subdivision represented the cellular division of an element. This cellular subdivision was ideally conceived through the same material, but each triangular cell had different levels of translucency (three different levels would be needed), producing a visual multilayer effect. Interspersed in the façade, combinations of triangular cells rotate, thus opening a visual and physical channel between internal and external environments. These openings were designed to produce movement in two different directions, one diagonally, intended to block western solar radiation when opened, and another horizontally, blocking the high southern solar rays. The translucent triangular elements follow the shading comfort diagram of the façade from the Ladybug analysis. Darker translucent triangular elements exist in more critical shading areas, medium-dark in regular shading areas, and lighter elements in parts with low demand for shading areas.

### 3.5. Group E (Filipa Osório and Pedro Frutuoso)

Group E began their project studying the sunrays of the southern façade during summer and winter solstices. Their selected functions were the ability to control entry and blockage of solar radiation as well as indoor glare. To achieve these functions, the selected actions were permeability, reflection, intersection and material. The group’s action > agent strategy was based on material translucency, a structure capable of producing intersection and reflection and a pattern with an adequate density. Their selected memes were the deciduous plants, whorl, bark trees, diaheliotropism and the nyctinastic movements. The correspondent Biomeme (Table 8) aimed at controlling not only solar radiation as well as the glare effect, using dynamic strategies, through the implantation of behavioral adaptations. The shading system has a random pattern, made of flexible materials, in order to enable a bidirectional movement.

The designed system is composed of a diamond-like mesh, which has its main axes aligned with the solstices’ solar altitudes (Figure 9). Each shading system unit corresponds to a single diamond capable of performing two perpendicular movements (Figure 10). The idea is that each diamond performs its movement independently from its neighbors, responding according to its individual sensor reading.

## 4. PoC 1.0 Evaluation and Discussion

The experience reported in this paper aimed at evaluating three important operative aspects of a new method intended at designing bioshading systems [7]: (1) the methodological clarity; (2) the experience sessions themselves (PoC 1.0 sessions); and (3) the methodological operability and applicability. The first two aspects were evaluated by the participants, while the third aspect was evaluated by the researchers based on the results from the five groups. The evaluation was based on three multi-criteria decision analysis (MCDA) assessment models using the MACBETH method [10] and the M-MACBETH software. MACBETH (Measuring Attractiveness by the Categorical Based Evaluation Technique) is an interactive approach that only requires qualitative judgments about differences in attractiveness to help a decision-maker or a decision support group to quantify the relative attractiveness of the options. As the judgments are introduced, their consistency is automatically checked. As output, a numeric scale is generated, entirely consistent with all judgments of the decision-maker. The weights attributed to the criteria were generated through a similar process. MACBETH was chosen because of its ability to incorporate various types of information (qualitative and quantitative) built through pairwise comparison judgments. In this way, the evaluation model can be shaped, in order to match the several types of analysis, through a co-participative decision-making process [11].

### 4.1. Method’s Clarity (Participants’ Evaluation)

For the method’s clarity, five evaluation criteria were defined, being evaluated on a scale from 1 to 5:Identify/analyze: Participants were questioned if the theoretical material introducing the method (BSDM), and the provided digital resources (analysis diagrams and three-dimensional modeling) were adequate for the general contextualization of the challenge and for the comprehension of the tasks;Diagnose: The participants were asked how easy or difficult they found to be the process of identifying and defining functions, actions and agents applied to the shading system;Discover/Explore: these criteria aimed at evaluating how easy/difficult was the Discover and Explore phases through the plants available and the surveys supplied;Conceptualize: how easy/difficult was the Biomeme creation process;Generate: How much the previous phases influenced/helped in the design process of the shading system.

The ranking given to each criterion (from 1 to 5) is translated to a percentage scale ranging from −62.5% to 100%, considering that 0% equals to 3 (3 was considered the acceptable minimum of success, below which the assessment is considered as negative). A weight of 20% was attributed to each criterion in order to achieve a final weighted average. At the end of the PoC 1.0 sessions, each participant was asked to evaluate the experience, assigning a value from 1 to 5 (where 1 was very weak, and 5 was very good) to each of the criteria. The results are shown in the ranking table (Table 9).

From the previous table, it is clear that the most fragile phase of the method was the Diagnose. During the evaluation session, participants were asked why it was so difficult for them to understand and accomplish the Diagnose phase (Architecture domain). The unanimous response was the lack of a diagram that explained how they should link the relations Functions > Actions > Agents, extracting the most relevant elements in order to proceed to the following phases. Some of the participants also pointed out that they struggled to extract the most relevant Functions that, à posteriori, should link to the Discover/Exploration phases of the Memes, suggesting that, instead, it would be more intuitive the connection of the Actions. Another participants’ input was related to the completion of the Biomimetic meme path matrix. It was suggested that a scheme relating the Memes events table with the Biomimetic Meme path matrix would increase the efficiency of the Biomeme creation avoinding spending non-creative time during the task. Generation was also considered a weaker phase due to the devoted period during the PoC 1.0 session, but its results will be more detailed in the PoC Session participants’ evaluation report. Identification and Analysis phases (Architectural domain), as well as Discover and Exploration phases (Nature domain) were highly punctuated, and no necessary alteration or improvement has been pointed out. Overall, the participants’ evaluation was strongly positive (Figure 11).

### 4.2. Evaluation of PoC Sessions (Participants’ Evaluation)

PoC Sessions participants’ evaluation was carried out using three assessment criteria:Clarity presenting the method: the participants were asked if, in an overall view, the presented method, its phases and tasks were presented in a clear and comprehensive way;Time for the session: in this criterion, the participants were asked if the time for the PoC sessions (total: 8h00) was adequate;Available means: this criterion aimed at evaluating the physical resources available for the PoC Sessions—room, computers and software.

The ranking given to each criterion was similar to the Method Clarity, from 1 to 5 (where 1 corresponds very weak, and 5 to very good), but in this case translated to a percentage scale ranging from −100% to 100%, again considering that 0% was equal to 3 in the sense that an assessment below 3 is equivalent to a negative performance. Reflecting the fact that clarity presenting the method is prevalent in influencing the participants’ experience and outputs, different weights were attributed to the three criteria: 46% to the Clarity Presenting the Method, and 27% to both criteria Time and Means. The results are in the PoC Session ranking table (Table 10).

During the PoC Session evaluation, Clarity Presenting the Method was the most well-scored criterion. However, as previously reported, participants felt a lack of procedural diagrams that could explain the connection Function > Action > Agent. Means were also pointed out as limited. Time was the poorest punctuated criterion. Participants reported that more time was needed to the Generate phase. Again, and overall, as it can be confirmed in Figure 12, participants’ evaluation of PoC sessions was positive.

### 4.3. Outputs and Results (Researchers’ Evaluation)

In the last two decades several methods and methodologies to implement biomimetic processes have been developed [12]. At the architectural level, the carried work remains, mostly on a theoretical level, with no practical application. The Biomimicry Design Spiral from Carl Hastrich [13], and the complex BioTriz developed by Nikolay and Olga Bogatyreva, based on the TRIZ—Teoriya Resheniya Izobretatelskikh Zadach methods [14] are references on the field of biomimetic application in architecture. However, the specialized lexicon, the generality of the stages as well as the lack of script for the passage of concept between domains, place these methodologies in the theoretical philosophical field of architecture. In 2014, Badarnah develops the BioGen [15] and in the same year Garcia-Holguera publish for the first time the Ecomimetic Design Method. The BioGen is defined as a biomimetic design concept generation methodology. The methodology is based on the principle of extraction of characteristics present in various types of ecosystems and their implementation in design rules. The Ecomimetic method based its conception on the Design Spiral, guiding its users since the abstraction, the transfer between ecosystems until the creation of the virtual model and its performance evaluation [16]. Despite the enormous contribution of previous works in the implementation of biomimetics in the design processes, there is still a lack of methods and methodologies that could guide the users during the transferring characteristics processes between the domains of nature and architecture [17]. In BSDM we work on this phase with the provision of several base tables that list events, describe them and still suggest possible interpretations, exemplifying and providing a working basis for the user of the methodology to proceed with its own transference and creation. In addition, BSDM provides its users with guides for the transition from the virtual model and analysis to its creation and production.

In order to evaluate the success of the method, the outputs and results obtained with this proof of concept, i.e., the projects produced by the participants, were assessed by the team of researchers. Two dimensions were considered: operability of the method and produced outputs. These two dimensions based the four assessment criteria for this evaluation:Method Clarity: related to Operability, this criterion aimed at evaluating, from the perspective of whom developed it, how was the method understood and conducted by the participants;Goal definition: related to Outputs, the objective of this criterion was to evaluate the level of coherence and understanding of the phases regarding the Architecture domain, as well as the definition of the functions > actions > agents relationship;Biomeme: also in the dimension of Outputs, this criterion aimed at evaluating the ability of abstraction, logical and deductive reasoning, as well as the creative individuality within the Nature domain;Technical implementation: to evaluate the degree of emulation of the design project, this Output-related criterion considered the implementation of technical and performance features.

Each of the above criteria contributed with 25% for the final weighted score. In the case of dimension Outputs, and in order to produce a detailed evaluation, the result of each group was assessed separately. Besides, each criterion (Goal definition, Biomeme and Technical implementation) is divided in assessment categories punctuated separately. The score of each criterion, as well as the Outputs global appreciation, is determined through the average of the partial scores (given to the different categories or calculated for each criterion). This intermediate assessment is detailed below.

Group A had a linear path. The pair soon understood the logic of the Goal’s definition and the creation process of the Biomeme, although, during the exploration phase, their search was limited, which then reflected in the abstraction period of the Nature domain. This limitation was transported to the Artifact domain, where the group struggled to design their fragmented ideas. The project was only developed in the ‘conceptual and formal’ plans, disregarding fabrication, or technical implementation. Even considering that it would be an open/close three-dimensional structure, the idea of how the mechanisms could be assembled and operated in the whole system would be essential for a more defined design project. Group A’s output evaluation is shown in Figure 13.

Group B had a difficult start-up; the shading system Goal definition was their most arduous task. However, after the task was completed, the subsequent phases belonging to the Nature domain provided them an engaging and productive creative process. Discover and Exploration phases were appropriately conducted, and the creation of the Biomeme resulted from an outstanding abstraction capacity. During the Generate phase, the group was able to design a primary mechanism, expressed in some notes about movements and material, but still, the global functioning of the system and its effect was unclear, as well as its mechanical system. Their complete evaluation can be checked in Figure 14.

Group C developed one of the most impressive projects of the PoC 1.0. During the Goal definition phase, the group followed the instructions and achieved an adequate result. Proceeding to the Nature domain, the group produced one of the most accurate researches, studying events and conceiving an astute strategy that enabled them, at this early stage, to connect their results to the Generate phase. This designing strategy enabled them to more rapidly connect their Biomeme to an innovative mechanism that sustained the design concept. In this way, the technical implementation did not come as an à posteriori design solution; instead it gave rise to it. As a result, the shading system proposal ensured an accurate challenge definition by framing the architecture challenge and defining its goals, a solid abstract biomimetic connection through the study of plants adaptations events, and an optimized technical implementation, providing the fundamental clues to its physical construction (Figure 15).

Group D’s shading design proposal initiated its Architectural domain based on a growing scale. The lack of some fundamental knowledge related to building climatic analysis was at the base of their difficulties. However, during the deductive process, information was linked and the relationship functions > actions > agents was appropriately defined. Nature domain phases were conducted successfully, leading to a coherent and adequate Biomeme. Their entire shading system proposal was designed through parametric tools, which led them to the conceptual idea of a possible technical implementation, by designing an intention of motion. The complete Output evaluation is shown in Figure 16.

Potentially due to the participants’ more extensive experience, Group E had no difficulties during the entire process of the new proposed method (BSDM). A correct and efficient response to the Architecture domain phases empowered them for the Nature domain, which was skillfully worked to accelerate the process to the Generate phase. During this phase, the group integrated the Goal definition constraints into a parametric model and designed a morphological proposal. The proposal offered not only a formal solution but also a structural and motion solution. Output evaluation can be checked in Figure 17.

The five groups presented each a shading system design proposal, based on environmental and climatic analysis, exploring plant adaptation events, studying and abstracting its features, considering motion hypothesis and mechanical implementation. Table 11 presents the overall evaluation integrating the Outputs dimension with the Method clarity dimension. Based on our assessment scale, two of the projects are considered as a negative output (Outputs global appreciation below 3) (Figure 18). Group A’s design proposal lacked ideas for technical implementation; and Group B had unclear goals and, similarly to Group A, lacked structural and technical implementation strategies. Based on the overall participation, the final evaluation of the projects reflects a successful experience.

## 5. Conclusions

PoC 1.0 provided diverse and valuable information regarding the Bioshading System Design Method, mainly concerning its application procedure and digital tool kit. Therefore, the evaluation from the PoC 1.0 experience and from the participants’ and researchers’ assessment was valid in leading to the identification of the most important aspects to be improved in the method. Overall, the method proved to be well structured and to lead its users through a design process of effective and creative façade shading systems. Some components of the method, as well as complementary material have to be (re)created:During the Architectural domain, besides providing lists with the primary functions, actions and agents, the Diagnose phase has to include an explanation of those items and to showcase them, by exemplifying possible correlations. To explain the different functions, actions and agents, a glossary should be created and made available to the method users. In order to exemplify the above-mentioned correlations, an illustrative diagram should also be delivered;Regarding the Nature domain, a diagram linking the Meme events table (Exploration phase) to the Biomimetic Meme path matrix (Conceptualize phase) would improve the users’ efficiency and optimize creative time during the process, accelerating the Biomeme creation;The Artifact domain requires time. PoC 1.0 initially aimed at achieving the Simulation phase during the second session. However, PoC 1.0 participants used the Simulation assigned time (one hour) for the Generate phase. Still, the projects designed during this experiment were sustained by a context and climatic analysis, what makes them more efficient and responsive to their surrounding environment context. Another important note about this phase is related to the high relevance of the users’ experience. Skilled digital fabrication users more easily integrate technical information in their designs, as well as skilled parametric designers more easily design motion concepts.

The proof of concept reported in this paper entirely covered the architecture conceptual process. From the challenge definition, through analysis and diagnose, to the abstraction phase through the conceptualization of a biomimetic meme. PoC 1.0 also validated the BSDM application at academic and professional levels. The present state of BSDM provides a limited selection of representative plant adaptation events based on a limited number of examples. To produce a more elaborated and complete database, a more extensive collaboration with interdisciplinary experts is required. Considering a more extensive database and since nature is always evolving and updating, it will be necessary to update constantly.

## Figures and Tables

**Figure 1 biomimetics-06-00008-f001:**
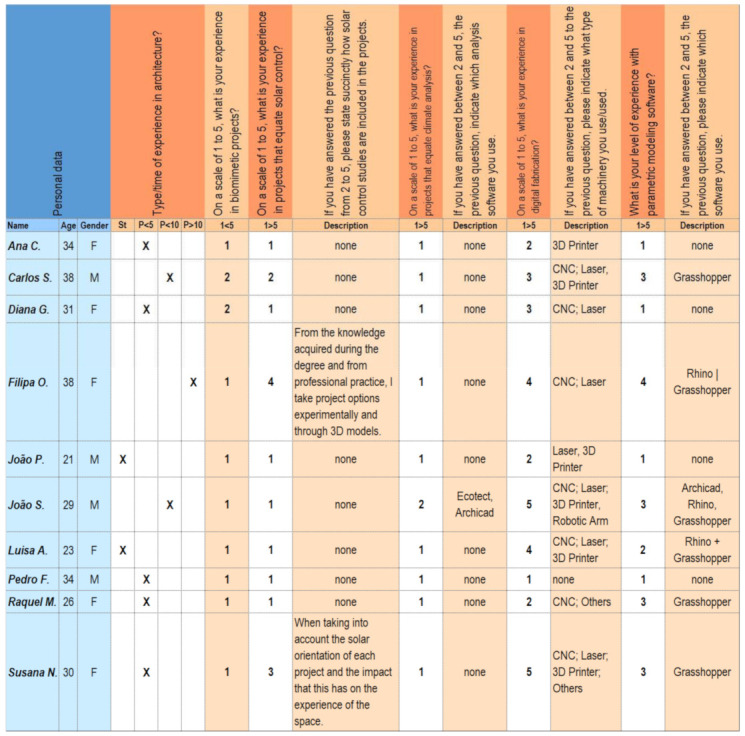
PoC 1.0—Participants data.

**Figure 2 biomimetics-06-00008-f002:**
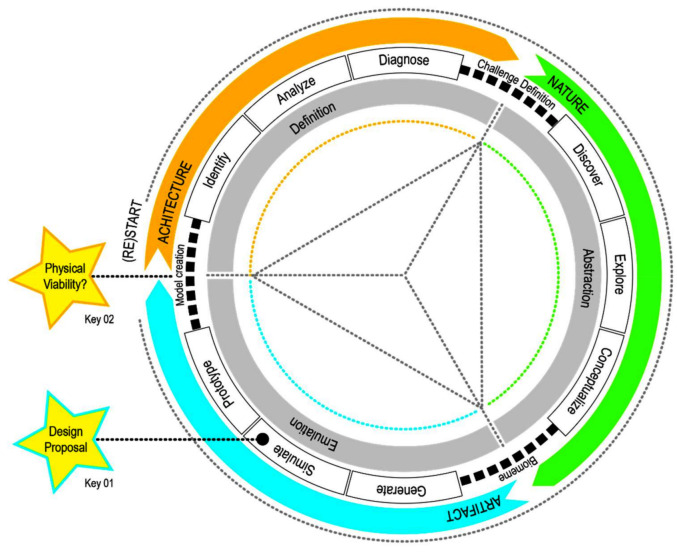
Bio-Shading Concept Design Method.

**Figure 3 biomimetics-06-00008-f003:**
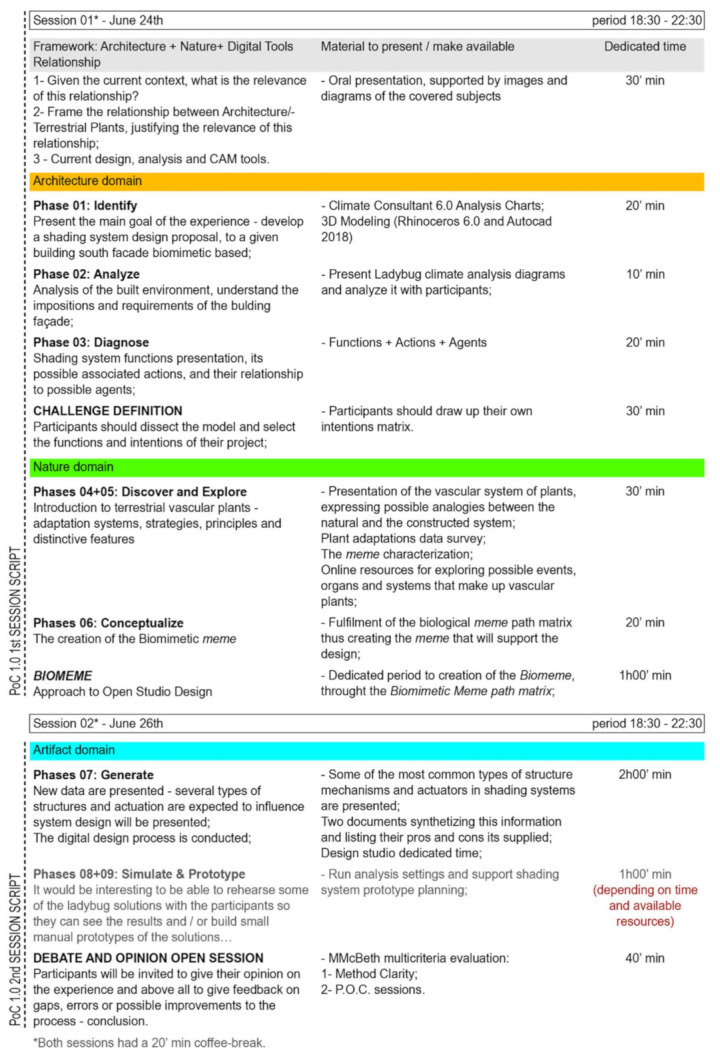
PoC 1.0—time script.

**Figure 4 biomimetics-06-00008-f004:**
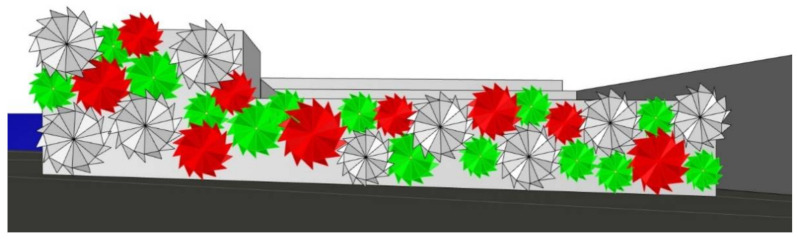
Group A—PoC project: Shading system design proposal.

**Figure 5 biomimetics-06-00008-f005:**
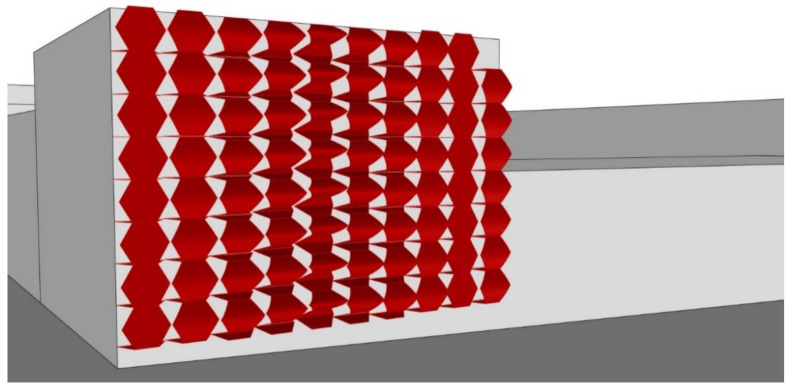
Group B–PoC project: Shading system design proposal.

**Figure 6 biomimetics-06-00008-f006:**
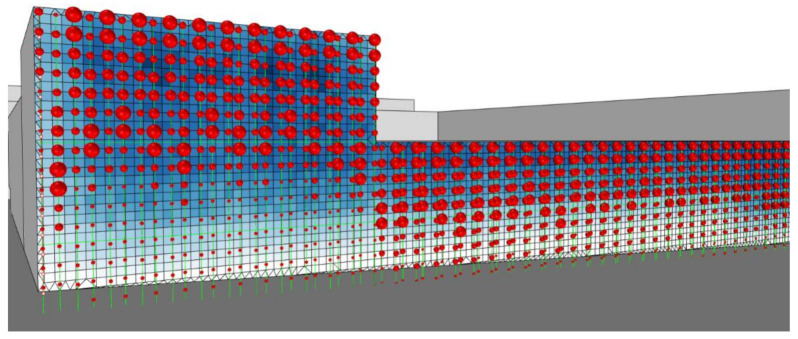
Group C—PoC project: Shading system design proposal.

**Figure 7 biomimetics-06-00008-f007:**
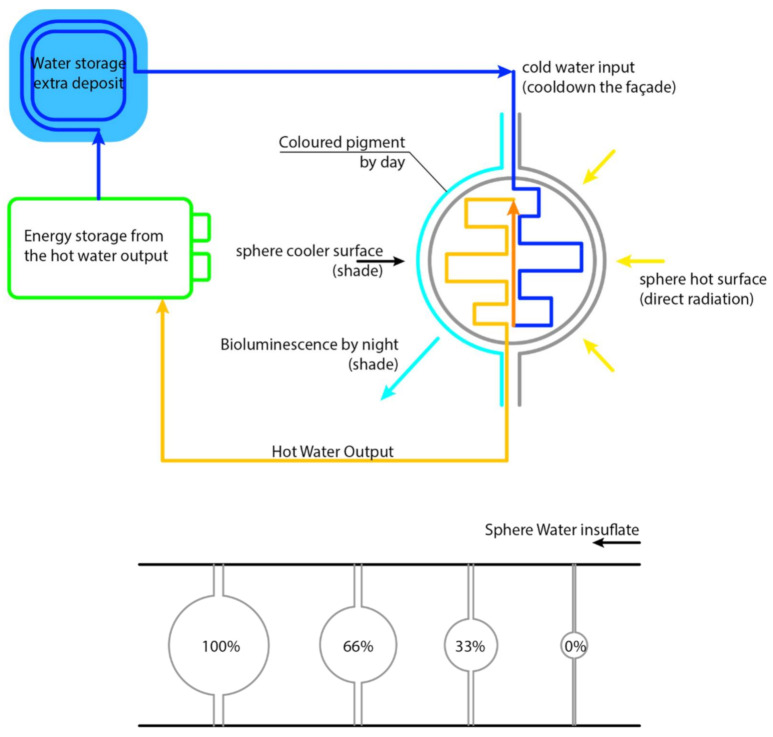
Group C—Shading system functional diagram.

**Figure 8 biomimetics-06-00008-f008:**
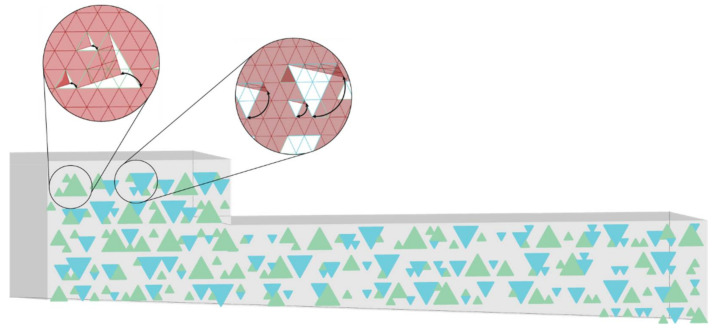
Group D—PoC project: Shading system design proposal.

**Figure 9 biomimetics-06-00008-f009:**
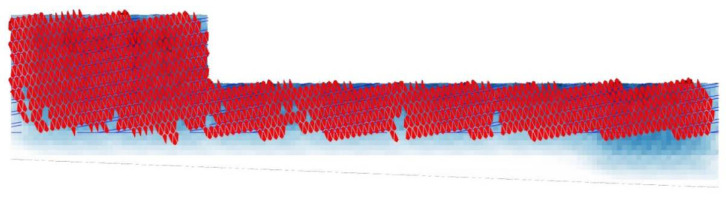
Group E—PoC project: Shading system design proposal.

**Figure 10 biomimetics-06-00008-f010:**
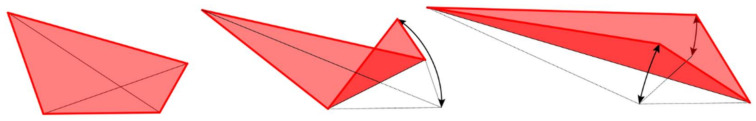
Group E—PoC project: diamond perpendicular movements.

**Figure 11 biomimetics-06-00008-f011:**
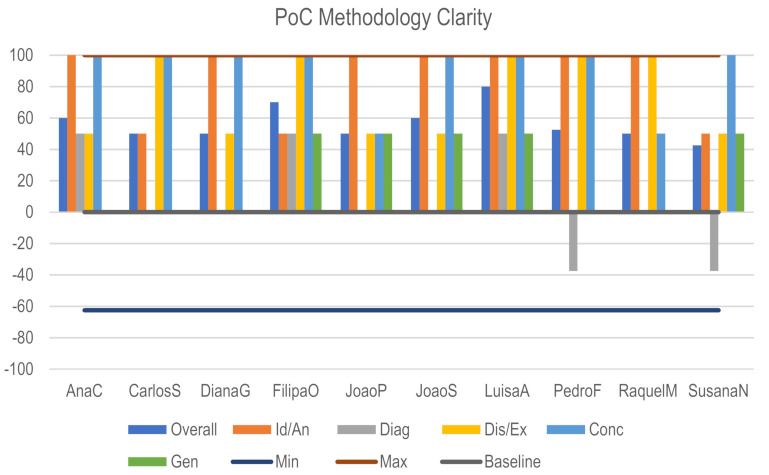
PoC Method Clarity.

**Figure 12 biomimetics-06-00008-f012:**
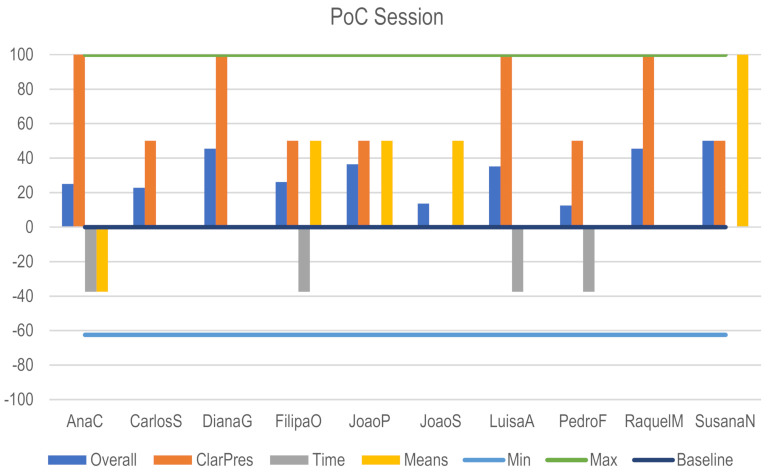
PoC session.

**Figure 13 biomimetics-06-00008-f013:**
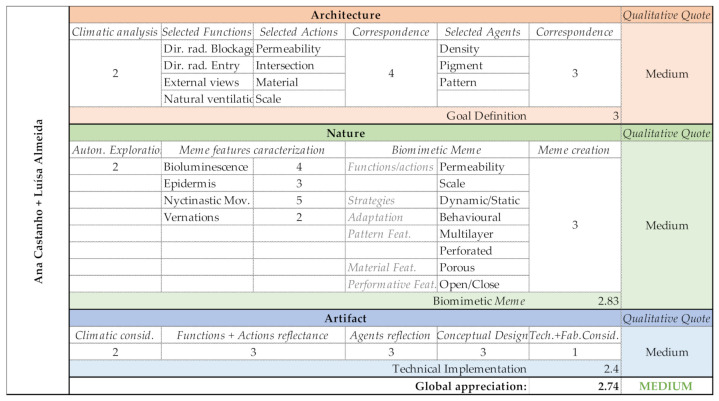
Group A—Output evaluation.

**Figure 14 biomimetics-06-00008-f014:**
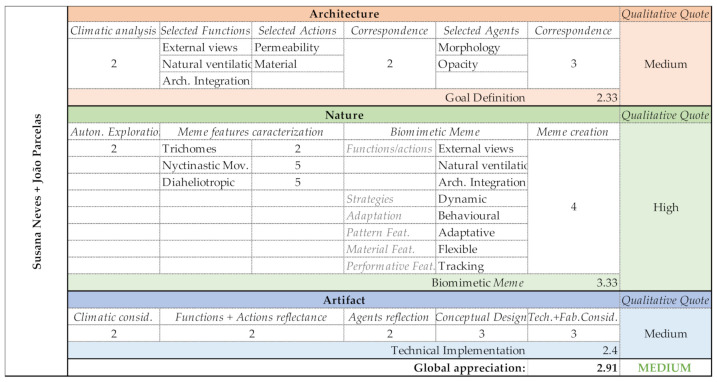
Group B—Output evaluation.

**Figure 15 biomimetics-06-00008-f015:**
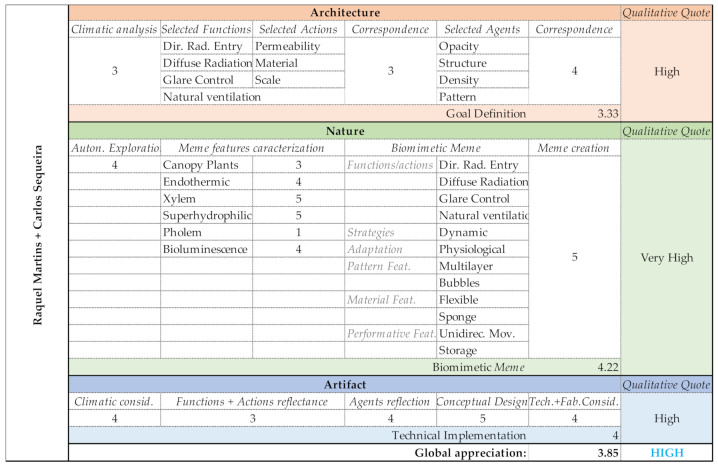
Group C—Output evaluation.

**Figure 16 biomimetics-06-00008-f016:**
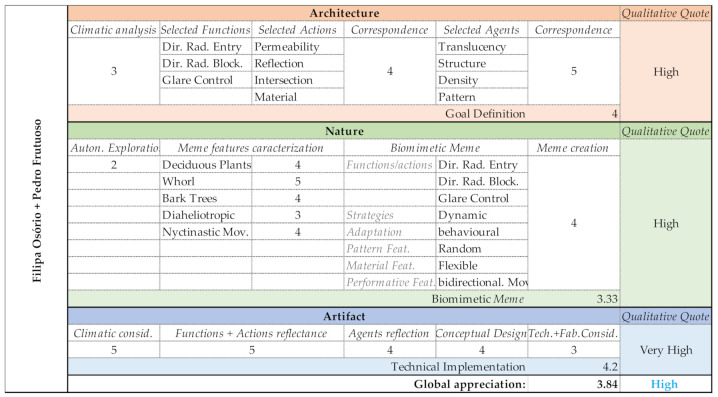
Group D—Output evaluation.

**Figure 17 biomimetics-06-00008-f017:**
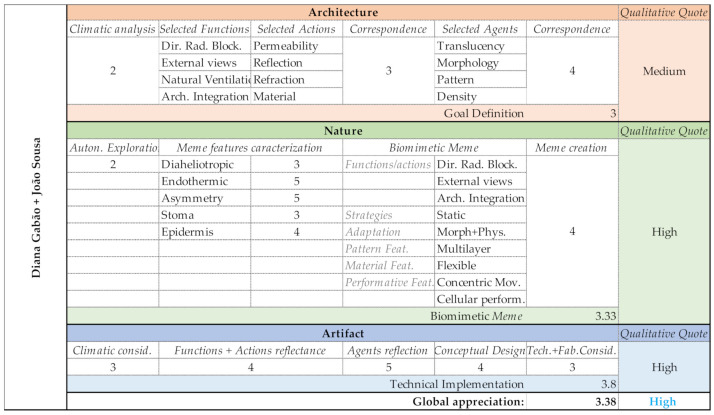
Group E—Output evaluation.

**Figure 18 biomimetics-06-00008-f018:**
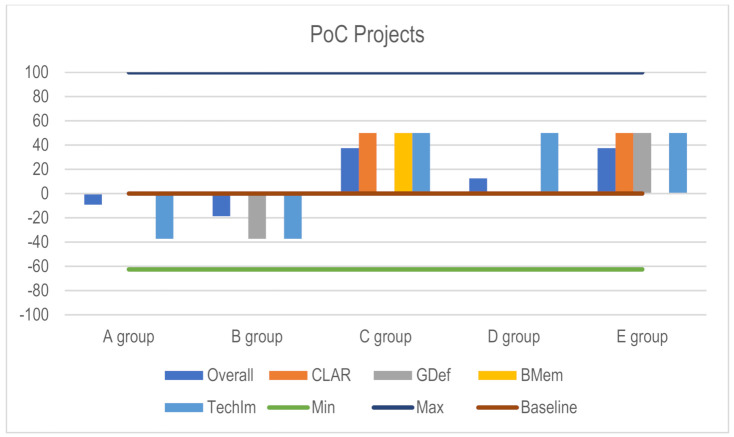
PoC 1.0 Projects—Final evaluation (when the assessment bar is not shown it means that the score corresponds to 3, lying in the graphic representation on the Baseline).

**Table 1 biomimetics-06-00008-t001:** Functions > Actions > Agents.

	Functions				Actions		Agents
1	Dir. Rad. entry			A	Permeability	a	Translucency
2	Dir. Rad. blockage			B	Reflection	b	Opacity
3	Diffuse Radiation			C	Refraction	c	Morphology
4	Glare control			D	Intersection	d	Structure
5	External views			E	Material	e	Density
6	Natural ventilation			F	Scale	f	Pigment
7	Architectural integration			G	…	g	Pattern
8	Others			H	…	h	Orientation
				I	…	i	Roughness
				J	…	j	Air flow
						k	…
						l	…
						m	…

**Table 2 biomimetics-06-00008-t002:** Meme Events example.

Meme Event	Adaptation	Strategy	Main Principles	Main Features
Bioluminescence	Behavioral	Dynamic	Occurs through a chemical reaction that produces light energy within an organism’s body.	Photosensitive
Epidermis	Physiological	Static	Epidermis is a layer of cells that covers the leaves, flowers, roots and stems of plants, forming a boundary between the plant and the external environment.	Multi-layer
Nyctinastic movements	Behavioral	Dynamic	The leaves of plants respond to the daily alternation between light and darkness by moving up and down.	Movement, open-close,
Vernation	Behavioral	Static	How the leaves are arranged on the buds, folding or curling.	Pattern

**Table 3 biomimetics-06-00008-t003:** Biomimetic Meme path matrix example provided to the PoC 1.0 participants.

Selected Functions	A Meme	B Meme	C Meme	Meme	Biomeme
1	X				X
2		X			X
3			X		X
4				X	X
Meme Strategies					
Dynamic	X	X		X	X
Static			X		
Meme Adaptation					
Morphological	X		X		X
Physiological				X	
Behavioral		X			
Meme Pattern features					
xxx	X				
xxx			X	X	X
Meme Material features					
xxx	X	X			X
xxx			X	X	X
Meme Performance Features					
xxx	X		X	X	X
xxx		X			

**Table 4 biomimetics-06-00008-t004:** Group A’s Biomeme.

**Group A—Biomimetic Meme**	
Functions/actions	Permeability
	Scale
Strategies	Dynamic/Static
Adaptation	Behavioral
Pattern Features	Multilayer
	Perforated
Material Features	Porous
Performance Features	Open/Close

**Table 5 biomimetics-06-00008-t005:** Group B’s Biomeme.

Group B—Biomimetic Meme	
Functions/actions	External views
	Natural ventilation
	Arch. Integration
Strategies	Dynamic
Adaptation	Behavioral
Pattern Features	Adaptative
Material Features	Flexible
Performance Features	Tracking

**Table 6 biomimetics-06-00008-t006:** Group C’s Biomeme.

Group C—Biomimetic Meme	
Functions/actions	Dir. Rad. Entry
	Diffuse Radiation
	Glare Control
	Natural ventilation
Strategies	Dynamic
Adaptation	Physiological
Pattern Features	Multilayer
	Bubbles
Material Features	Flexible
	Sponge
Performance Features	Unidirectional Mov.
	Storage

**Table 7 biomimetics-06-00008-t007:** Group D’s Biomimetic Meme.

Group D—Biomimetic Meme	
Functions/actions	Dir. Rad. Block.
	External views
	Arch. Integration
Strategies	Static
Adaptation	Morphological/Physiological
Pattern Features	Multilayer
Material Features	Hard
Performance Features	Concentric Movement
	Cellular performance

**Table 8 biomimetics-06-00008-t008:** Group E’s Biomimetic Meme.

Group E—Biomimetic Meme	
Functions/actions	Dir. Rad. Entry
	Dir. Rad. Block.
	Glare Control
Strategies	Dynamic
Adaptation	Behavioral
Pattern Features	Random
Material Features	Flexible
Performance Features	Bidirectional. Mov.

**Table 9 biomimetics-06-00008-t009:** Method’s Clarity ranking Table 1.

Method Clarity—Ranking Table				
Id/An (20%)	Diag (20%)	Dis/Ex (20%)	Conc (20%)	Gen (20%)
Id/An 5	Diag 5	Dis/Ex 5	Conc 5	Gen 5
AnaC	AnaC	CarlosS	AnaC	FilipaO
DianaG	FilipaO	FilipaO	CarlosS	JoãoP
JoãoP	LuísaA	LuísaA	DianaG	JoãoS
JoãoS	Diag 3	PedroF	FilipaO	LuísaA
LuísaA	CarlosS	RaquelM	JoãoS	SusanaN
PedroF	DianaG	AnaC	LuísaA	Gen 3
RaquelM	JoãoP	DianaG	PedroF	AnaC
CarlosS	JoãoS	JoãoP	SusanaN	CarlosS
FilipaO	RaquelM	JoãoS	JoãoP	DianaG
SusanaN	PedroF	SusanaN	RaquelM	PedroF
Id/An 3	SusanaN	Dis/Ex 3	Conc 3	RaquelM

Ranking Table The name of each participant has been abbreviated: AnaC (Ana Castanho); CarlosS (Carlos Sequeira); DianaG (Diana Gabão); FilipaO (Filipa Osório); JoãoP (João Parcelas); JoãoS (João Sousa); LuísaA (Luísa Almeida); PedroF (Pedro Frutuoso); RaquelM (Raquel Martins); SusanaN (Susana Neves). Legend: 
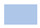
 5 (Very good); 
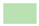
 4 (Good); 
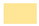
 3 (Moderate/minimum acceptable); 
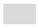
 2 (Weak); 

 1 (Very Weak). Abbreviations: Diag—Diagnose; Dis/Ex—Discover and Exploration; Conc—Conceptualize; Gen—Generate; Id/An—Identification and Analysis.

**Table 10 biomimetics-06-00008-t010:** PoC session—Ranking Table.

PoC Session—Ranking Table		
ClarPres (45.46%)	Time (27.27%)	Means (27.27%)
ClarPres 5	Time5	Means5
AnaC	Time3	SusanaN
DianaG	CarlosS	FilipaO
LuísaA	DianaG	JoãoP
RaquelM	JoãoP	JoãoS
CarlosS	JoãoS	Means3
FilipaO	RaquelM	CarlosS
JoãoP	SusanaN	DianaG
PedroF	AnaC	LuisaA
SusanaN	FilipaO	PedroF
ClarPres3	LuísaA	RaquelM
JoãoS	PedroF	AnaC

Legend: 
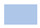
 5 (Very good); 
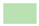
 4 (Good); 
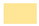
 3 (Moderate/minimum acceptable); 
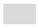
 2 (Weak); 

 1 (Very Weak). Abbreviations: ClarPres—Clarity Presenting Method; Time—Time; Means—Means.

**Table 11 biomimetics-06-00008-t011:** PoC 1.0 Projects—Ranking Table.

Ranking Table			
Clar (25%)	GDef (25%)	BMem (25%)	TechIm (25%)
Clar5	GDef5	BioMem5	TechImpl5
Group C	Group E	Group C	Group C
Group E	GDef3	BioMem3	Group D
Clar3	Group A	Group A	Group E
Group A	Group C	Group B	TechImpl3
Group B	Group D	Group D	Group A
Group D	Group B	Group E	Group B

Legend: 
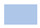
 5 (Very good); 
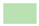
 4 (Good); 
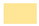
 3 (Moderate/minimum acceptable); 
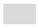
 2 (Weak); 

 1 (Very Weak). Abbreviations: Clar—Method Clarity; GDef—Goal Definition; BMem- Biomimetic Meme; TechIm—Technical Implementation.

## Data Availability

The data presented in this study are openly available in ISCTE-IUL repositorium at http://hdl.handle.net/10071/20600; Thesis identifier: 101564821; ISBN: 978-989-781-331-3.

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
