# Peer review of "Proof of Concept (PoC) 1.0—Implementing a Bioshading System Design Method"

_biomimetics, 2021, doi:10.3390/biomimetics6010008_

Round 1

Reviewer 1 Report

It can be said that this paper has a good topic originality, with enough relevance and importance. Also good in methodology, discussion and conclusion.

The topic is of real importance and concern for all design practitioners and academics. The manuscript is professionally written, easy to read, jargon-free and free from grammatical or spelling errors. The work is a sufficiently original contribution to its field.

The manuscript describes clear and well-justified the research, and positions the work stating the overall problem, while framing enough the investigation's theoretical approach.

Only needs to be corrected all the places where appears "Error! reference source not found".

Author Response

We thank you for your comments that flatter our work.

Regarding the error, where the message "Error! reference source not found" appeared due to the export of the word file to PDF, where the pdf lost the image and table references. We have already corrected it and found a way to prevent it from happening again.

A revised version of the article is attached.

Regards

Reviewer 2 Report

The article is written in rather a descriptive and abstract manner. Much more reflection and detail are necessary. Some tables would benefit from being transferred to diagrams. This would enable a better reflection. Table 1 is unreadable. Some terms, both internal and external, are lacking explanation. The paper would also benefit from much deeper explanations and illustrations of the natural systems that were used for the inspiration for the experiments. The discussion is written in a form of conclusion, whilst the discussion is missing. The discussion should compare the work with other authors in the field.

Author Response

The authors would first of all like to thank you for your valuable contributions and comments, which from our point of view have greatly enriched our work.

The whole narrative on the description of the method has been revised, which has led us to a restructuring and reconnection between the body of text and the more descriptive and detailed tables.
This more descriptive and detailed narrative also helped us with the explanation and description to the reader about the internal and external terms of the method.

Table 1, which in fact contains a lot of information, was restructured in order to compile several cells and the numerical scale was used to simplify and optimise its area.

A clear separation has been created between what is discussion and what is conclusion.

In order to be able to check all the changes made, you will find attached the word document with the active Microsoft Word "Track Changes".

(Please, also see the attachment.)
